# Adaptive Path-Integral Autoencoder: Representation Learning and Planning for Dynamical Systems

**Jung-Su Ha, Young-Jin Park, Hyeok-Joo Chae, Soon-Seo Park, and Han-Lim Choi**
Department of Aerospace Engineering & KI for Robotics, KAIST
Daejeon 305-701, Republic of Korea
{{jsha, yjpark, hjchae, sspark}@lics., hanlimc@}kaist.ac.kr

## Abstract

We present a representation learning algorithm that learns a low-dimensional latent dynamical system from high-dimensional *sequential* raw data, e.g., video. The framework builds upon recent advances in amortized inference methods that use both an inference network and a refinement procedure to output samples from a variational distribution given an observation sequence, and takes advantage of the duality between control and inference to approximately solve the intractable inference problem using the path integral control approach. The learned dynamical model can be used to predict and plan the future states; we also present the efficient planning method that exploits the learned low-dimensional latent dynamics. Numerical experiments show that the proposed path-integral control based variational inference method leads to tighter lower bounds in statistical model learning of sequential data. The supplementary video[1] and the implementation code[2] are available online.

## 1  Introduction

Unsupervised learning of the underlying dynamics of sequential high-dimensional sensory inputs is the essence of intelligence, because the agent should utilize the learned dynamical model to predict and plan the future state. Such learning problems are formulated as latent or generative model learning assuming that observations were emerged from the low-dimensional latent states, which includes an intractable posterior inference of latent states for given input data. In the amortized variational inference framework, an inference network is introduced to output variational parameters of an approximate posterior distribution. This allows for a fast approximate inference procedure and efficient end-to-end training of the generative and inference networks when the learning signals from a loss function are back-propagated into the inference network with reparameterization trick [Kingma and Welling, 2014, Rezende et al., 2014]. The learning procedure is based on optimization of a surrogate loss, a lower-bound of data likelihood, which results in two source of sub-optimality: an approximation gap and an amortization gap [Krishnan et al., 2018, Cremer et al., 2018]; the former comes from the sub-optimality of variational approximation (the gap between true posterior and optimal variational distribution) and the latter is caused by the amortized approximation (the gap between the optimal variational distribution and the distribution from the inference network). Recently, several works, e.g., [Hjelm et al., 2016, Krishnan et al., 2018, Kim et al., 2018], combined iterative refinement procedures with the amortized inference, where the output distribution of the inference network is used as a warm-start point of refinement. This technique is referred to as the *semi-amortized* inference and, since refined variational distributions do not rely only on the inference network, the sub-optimality from amortization gap can be mitigated.

For sequential data modeling, a generative model should be considered as a dynamical system and a more sophisticated (approximate) inference method is required. With the assumption that the underlying dynamics has the Markov property, a state space model can be introduced and it allows the inference network to be structured so as to mimic the factorized form of a true posterior distribution [Krishnan et al., 2017, Karl et al., 2017, Fraccaro et al., 2017]. The efficient end-to-end training with the amortized inference is also possible here, where the inference network should output the variational distribution of latent state trajectories for given observation sequences. Even when the inference network is structured, the amortization gap increases inevitably because the inference should be performed in the *trajectory* space.

In this work, we present a semi-amortized variational inference method operated in the trajectory space. For a generative model given by a state space model, an initial state distribution and control inputs serve as parameters of variational distributions; the inference network is trained to output these variational parameters such that the corresponding latent trajectory well-describes the observation sequence. In this certain formulation, the divergence between the prior and the variational distribution is naturally derived from stochastic calculus and then the inference problem can be converted into a stochastic optimal control (SOC) problem, i.e., so-called control-inference duality [Todorov, 2008, Ruiz and Kappen, 2017]. In the SOC view, what the inference network does is to approximate the optimal control policy, which is hardly thought to be well-done when we observe that SOC problems are hard to solve at once, so iterative methods are generally used to solve the problems [Todorov, 2008, Tamar et al., 2016, Okada et al., 2017]. Thus, we adopt the adaptive path-integral control method to iteratively refine the variational parameters. We show that because samples from the refined variational distribution build tighter lower-bound and all the refinement procedures are differentiable, efficient end-to-end training is possible. Moreover, because the proposed framework is based on the SOC method, the same structure can be utilized to plan the future observation sequence, where the learned low-dimensional stochastic dynamics is used to explore the high-dimensional observation space efficiently.

## 2 Background

### 2.1 Statistical Modeling of Sequential Observations

Suppose that we have a set of observation sequences $\{\mathbf{x}_{1:K}^{(i)}\}_{i=1,\dots,I}$, where $\mathbf{x}_{1:K}^{(i)} \equiv \{\mathbf{x}_k; \forall k = 1,\dots,K\}^{(i)}$ are i.i.d. sequences of observation that lie on (possibly high-dimensional) data space, $\mathcal{X} \subset \mathbb{R}^{d_x}$. The problem of interest is to build a probabilistic model that explains the given observations well. If a model is parameterized with $\theta$, the problem is formulated as a maximum likelihood estimation (MLE) problem:

$$\theta^* = \operatorname*{argmax}_{\theta} \sum_i \log p_\theta(\mathbf{x}_{1:K}^{(i)}). \tag{1}$$

In this work, the observations are assumed to be emerged from a latent dynamical system, where a latent state trajectory, $\mathbf{z}_{[0,T]} \equiv \{\mathbf{z}(t); \forall t \in [0,T]\}$, lies on a (possibly low-dimensional) latent space, $\mathcal{Z} \subset \mathbb{R}^{d_z}$:

$$p_\theta(\mathbf{x}_{1:K}) = \int p_\theta(\mathbf{x}_{1:K}|\mathbf{z}_{[0,T]}) dp_\theta(\mathbf{z}_{[0,T]}), \tag{2}$$

where $p_\theta(\mathbf{x}_{1:K}|\mathbf{z}_{[0,T]})$ and $p_\theta(\mathbf{z}_{[0,T]})$ are called a conditional likelihood and a prior distribution, respectively[3] . In particular, we consider the state space model where latent states are governed by a continuous-time stochastic differential equation (SDE), i.e., the prior $p_\theta(\mathbf{z}_{[0,T]})$ is a probability measure of a following system:

$$d\mathbf{z}(t) = \mathbf{f}(\mathbf{z}(t))dt + \sigma(\mathbf{z}(t))d\mathbf{w}(t), \ \mathbf{z}(0) \sim p_0(\cdot), \tag{3}$$

where $\mathbf{w}(t)$ is a $d_u$-dimensional Wiener process. Additionally, a conditional likelihood of sequential observations is assumed to be factorized along the time axis:

$$p_\theta(\mathbf{x}_{1:K}|\mathbf{z}_{[0,T]}) = \prod_{k=1}^K p_\theta(\mathbf{x}_k|\mathbf{z}(t_k)), \tag{4}$$

where $\{t_k\}$ is a sequence of discrete time points with $t_1 = 0, t_K = T$.

## 2.2 Amortized Variational Inference and Multi-Sample Objectives

The objective function (1) cannot be optimized directly because it contains the intractable integration. To circumvent the intractable inference, a variational distribution $q(\cdot)$ is introduced and then a surrogate loss function $\mathcal{L}(q, \theta; \mathbf{x})$, which is called the evidence lower bound (ELBO), can be considered alternatively:

$$\log p_\theta(\mathbf{x}) = \log \int p_\theta(\mathbf{x}|\mathbf{z}) p_\theta(\mathbf{z}) d\mathbf{z} \geq \mathbb{E}_{q(\mathbf{z})} \left[ \log \frac{p_\theta(\mathbf{x}|\mathbf{z}) p_\theta(\mathbf{z})}{q(\mathbf{z})} \right] \equiv \mathcal{L}(q, \theta; \mathbf{x}), \qquad (5)$$

where $q(\cdot)$ can be any probabilistic distribution over $\mathcal{Z}$ of which support includes that of $p_\theta(\cdot)$. The gap between the log-likelihood and the ELBO is the Kullback–Leibler (KL) divergence between $q(\mathbf{z})$ and the posterior $p_\theta(\mathbf{z}|\mathbf{x})$:

$$\log p_\theta(\mathbf{x}) - \mathcal{L}(q, \theta; \mathbf{x}) = D_{KL}(q(\mathbf{z})||p_\theta(\mathbf{z}|\mathbf{x})). \qquad (6)$$

In particular, the *amortized variational inference* approach introduces a conditional variational distribution, $\mathbf{z} \sim q_\phi(\cdot|\mathbf{x})$, to approximate the intractable posterior distribution. The variational distribution $q_\phi(\cdot|\mathbf{x})$, which is referred to as the *inference network*, is parameterized by $\phi$, so $\theta$ and $\phi$ can be simultaneously updated with $\nabla_{(\theta,\phi)} \mathcal{L}(q_\phi, \theta; \mathbf{x})$ using the stochastic gradient ascent. Variational autoencoders (VAEs) [Kingma and Welling, 2014, Rezende et al., 2014] make $q_\phi(\cdot|\mathbf{x})$ a reparameterizable distribution, where $\mathbf{z} = g_\phi(\mathbf{x}, \epsilon)$ is a differentiable deterministic function of an observation $\mathbf{x}$ and $\epsilon \sim d(\cdot)$ sampled from a known base distribution $d(\cdot)$. Then, the gradient can be estimated as: $\nabla_{(\theta,\phi)} \mathcal{L}(q_\phi, \theta; \mathbf{x}) = \mathbb{E}_{d(\epsilon)} \left[ \nabla_{(\theta,\phi)} \log \frac{p_\theta(\mathbf{x}, g_\phi(\mathbf{x}, \epsilon))}{q_\phi(g_\phi(\mathbf{x}, \epsilon))} \right]$, which generally yields a low variance estimator.

A tighter lower bound is achieved by using multiple samples, $\mathbf{z}^{1:L}$, independently sampled from $q_\phi$:

$$\mathcal{L}^L \equiv \mathbb{E}_{\mathbf{z}^{1:L} \sim q_\phi(\cdot|\mathbf{x})} \left[ \log \frac{1}{L} \sum_{l=1}^{L} \frac{p_\theta(\mathbf{x}, \mathbf{z}^l)}{q_\phi(\mathbf{z}^l|\mathbf{x})} \right]. \qquad (7)$$

It is proven that, as $L$ increases, the bounds get tighter, i.e., $\log p_\theta(\mathbf{x}) \geq \cdots \geq \mathcal{L}^{L+1} \geq \mathcal{L}^L \geq \cdots$, and the gap eventually vanishes [Burda et al., 2016, Cremer et al., 2017].This multi-sample objective (7) is in the class of Monte Carlo objectives (MCO) in the sense that it utilizes independent samples to estimate the marginal likelihood [Mnih and Rezende, 2016], $\hat{p}_\theta(\mathbf{x}) = \frac{1}{L} \sum_{l=1}^{L} \frac{p_\theta(\mathbf{x}, \mathbf{z}^l)}{q_\phi(\mathbf{z}^l|\mathbf{x})}$, $\mathbf{z}^l \sim q_\phi(\cdot|\mathbf{x})$. Defining $w_{\theta,\phi}(\mathbf{x}, \mathbf{z}^l) \equiv \frac{p_\theta(\mathbf{x}, \mathbf{z}^l)}{q_\phi(\mathbf{z}^l|\mathbf{x})}$ and $\tilde{w}^l \equiv \frac{w_{\theta,\phi}(\mathbf{x}, \mathbf{z}^l)}{\sum_i w_{\theta,\phi}(\mathbf{x}, \mathbf{z}^i)}$, the gradient of (7) is given by:

$$\nabla_{(\theta,\phi)} \mathcal{L}^L = \mathbb{E}_{\epsilon^{1:L} \sim d(\cdot)} \left[ \sum_{l=1}^{L} \tilde{w}^l \nabla_{(\theta,\phi)} \log w_{\theta,\phi}(\mathbf{x}, g_\phi(\mathbf{x}, \epsilon^l)) \right]. \qquad (8)$$

Since the parameter update is averaged over multiple samples with the weights $\tilde{w}^l$, the above procedure is referred to as importance weighted autoencoders (IWAEs) [Burda et al., 2016]. The performance of IWAE's training crucially depends on the variance of the importance weights $\tilde{w}$ (or equivalently, on the effective sample size), which can be reduced by (i) increasing the number of samples and (ii) decreasing the gap between the proposal and the true posterior distribution; when the proposal $q_\phi(\cdot|\mathbf{x})$ is equal to the true posterior $p_\theta(\cdot|\mathbf{x})$, the variance is reduced to 0, i.e., $\tilde{w}^l = 1/L$.

## 2.3 Semi-Amortized Variational Inference with Iterative Refinement

As mentioned previously, the performance of generative model learning depends on the gap between the variational and the posterior distributions. Thus, the amortized inference has two sources of this gap: the approximation and amortization gaps [Krishnan et al., 2018, Cremer et al., 2018]. The approximation gap comes up by using the variational distribution to approximate the posterior distribution, which is given by the KL-divergence between the posterior distribution and the optimal variational distribution. The amortization gap is caused by the limit of the expressive power of inference networks, where the variational parameters are *not* individually optimized for each observation but amortized over entire observations. To address the issue of the amortization gap, a hybrid approach can be considered; for each observation, the variational distribution is refined individually from the output of the inference network. Compared to the amortized variational inference, this hybrid approach, coined semi-amortized variational inference, allows for utilizing better variational parameters in model learning.

# 3 Path Integral Adaptation for Variational Inference

## 3.1 Controlled SDE as variational distribution and structured inference network

When handling sequential observations, the variational distribution family should be carefully chosen so as to efficiently handle increasing dimensions of variables along the time-axis. In this work, the variational proposal distribution is given by the trajectory distribution of a controlled stochastic dynamical system, where the controls, $\mathbf{u} \in \mathbb{R}^{d_u}$, and parameters of an initial state distribution, $q_0$, serve as variational parameters, i.e., the proposal $q_\mathbf{u}(\mathbf{z}_{[0,T]})$ is a probability measure of a following system:

$$d\mathbf{z}(t) = \mathbf{f}(\mathbf{z}(t))dt + \sigma(\mathbf{z}(t))(\mathbf{u}(t)dt + d\mathbf{w}(t)), \ \mathbf{z}(0) \sim q_0(\cdot). \tag{9}$$

By applying Girsanov's theorem in Appendix A that provides the likelihood ratio between $p(\mathbf{z}_{[0,T]})$ and $q_\mathbf{u}(\mathbf{z}_{[0,T]})$, the ELBO is written as:

$$\mathcal{L} = \mathbb{E}_{q_\mathbf{u}(\mathbf{z}_{[0,T]})} \left[ \log p_\theta(\mathbf{x}_{1:K}|\mathbf{z}_{[0,T]}) + \log \frac{p_0(\mathbf{z}(0))}{q_0(\mathbf{z}(0))} - \frac{1}{2} \int_0^T ||\mathbf{u}(t)||^2 dt - \int_0^T \mathbf{u}(t)^T d\mathbf{w}(t) \right]. \tag{10}$$

Then, the problem of finding the optimal variational parameters $\mathbf{u}^*$ and $q_0^*$ (or equivalently, the best approximate posterior) can be formulated as a SOC problem:

$$\mathbf{u}^*, q_0^* = \underset{\mathbf{u}, q_0}{\operatorname{argmin}} \, \mathbb{E}_{q_\mathbf{u}(\mathbf{z}_{[0,T]})} \left[ V(\mathbf{z}_{[0,T]}) + \frac{1}{2} \int_0^T ||\mathbf{u}(t)||^2 dt + \int_0^T \mathbf{u}(t)^T d\mathbf{w}(t) \right], \qquad \textbf{(SOC)}$$

where $V(\mathbf{z}_{[0,T]}) \equiv -\log \frac{p_0(\mathbf{z}(0))}{q_0(\mathbf{z}(0))} - \sum_{k=1}^K \log p_\theta(\mathbf{x}_k|\mathbf{z}(t_k))$ serves as a state cost of the SOC problem.

Suppose that the control policy is discretized along the time-axis with the control parameters $\{\mathbf{u}_k^{ff}, \mathbf{K}_k\}_{k=1,...,K-1}$ as $\mathbf{u}(t, \mathbf{z}(t)) = \mathbf{u}_k^{ff} - \mathbf{K}_k \mathbf{z}(t)$, $\forall t \in [t_k, t_{k+1})$, and the initial distribution is modeled to be the Gaussian distribution, $q_0(\cdot) = \mathcal{N}(\cdot; \hat{\mu}_0, \hat{\Sigma}_0)$. Once the inference problem is converted into the SOC problem, the principle of optimality [Bellman, 2013] provides the sophisticated and efficient structure of inference networks. Note that, by the principle of optimality, the optimal initial state distribution depends on the cost for all time horizon $[0, T]$ but the optimal control policy at $t$ only relies on the future cost in $(t, T]$. Such a structure can be implemented using a backward recurrent neural network (RNN) to output the approximate optimal control policy; while the hidden states of the backward RNN compress the information of a given observation sequence backward in time, the hidden state at each time step, $k = K - 1, ..., 2$, outputs the control policy parameters, $\{\mathbf{u}_k^{ff}, \mathbf{K}_k\}$. Finally, the first hidden state additionally outputs the initial distribution parameters, $\{\hat{\mu}_0, \hat{\Sigma}_0, \mathbf{u}_1^{ff}, \mathbf{K}_1\}$. For the detailed descriptions and illustrations, see Fig. 3(a) and Algorithm 2 in Appendix C.

## 3.2 Adaptive Path-Integral Autoencoder

(**SOC**) is in a class of linearly-solvable optimal control problems [Todorov, 2009] of which the objective function can be written as a KL-divergence form:

$$J = D_{KL} \left( q_\mathbf{u}(\mathbf{z}_{[0,T]}) || p^*(\mathbf{z}_{[0,T]}) \right) - \log \xi, \tag{11}$$

where $p^*$, represented as $dp^*(\mathbf{z}_{[0,T]}) = \exp(-V(\mathbf{z}_{[0,T]}))dp_\theta(\mathbf{z}_{[0,T]})/\xi$, is a probability measure induced by optimally-controlled trajectories and $\xi \equiv \int \exp(-V(\mathbf{z}_{[0,T]}))dp_\theta(\mathbf{z}_{[0,T]})$ is a normalization constant (see Appendix A for details). By applying Girsanov's theorem again, the optimal trajectory distribution is expressed as:

$$dp^*(\mathbf{z}_{[0,T]}) \propto dq_\mathbf{u}(\mathbf{z}_{[0,T]}) \exp \left( -S_\mathbf{u}(\mathbf{z}_{[0,T]}) \right), \tag{12}$$

$$S_\mathbf{u}(\mathbf{z}_{[0,T]}) = V(\mathbf{z}_{[0,T]}) + \frac{1}{2} \int_0^T ||\mathbf{u}(t)||^2 dt + \int_0^T \mathbf{u}(t)^T d\mathbf{w}(t). \tag{13}$$

This implies that the optimal trajectory distribution can be approximated by sampling a set of trajectories according to the controlled dynamics with $\mathbf{u}(t)$, i.e. $\mathbf{z}_{[0,T]}^l \sim q_\mathbf{u}(\cdot)$, and assigning their

importance weights as $\tilde{w}^l = \frac{\exp(-S_{\mathbf{u}}(\mathbf{z}^l_{[0,T]}))}{\sum_{i=1}^L \exp(-S_{\mathbf{u}}(\mathbf{z}^i_{[0,T]}))}$, $\forall l \in \{1,...,L\}$. Similar to the MCO's case, the variance of importance weights decreases as the control input $\mathbf{u}(\cdot)$ gets closer to the true optimal control input $\mathbf{u}^*(\cdot)$ and it reduces to 0 when $\mathbf{u}(t) = \mathbf{u}^*(t, \mathbf{z}(t))$ [Thijssen and Kappen, 2015].

The path-Integral control is a sampling-based SOC method, which approximates the optimal trajectory distribution, $\hat{p}^*$, with weighted sample trajectories using (12)–(13) and updates control parameters based on moment matching of $q_{\mathbf{u}}$ to $\hat{p}^*$. Suppose that $\hat{p}^*$ is approximated with sample trajectories and their weights, $\{\mathbf{z}^l_{[0,T]}, \tilde{w}^l\}_{l=1,...,L}$, as above and let $\mathbf{u}^{ff}(t)$ and $\mathbf{K}(t)$ represent feedforward control and feedback gain, respectively. This work considers a standardized linear feedback controller to regularize the first and second moments of trajectory distributions, where a control input has a form as:

$$\mathbf{u}(t) = \mathbf{u}^{ff}(t) + \mathbf{K}(t)\Sigma^{-1/2}(t)(\mathbf{z}(t) - \mu(t)), \tag{14}$$

where $\mu(t) = \sum_{l=1}^L \tilde{w}^l \mathbf{z}^l(t)$ and $\Sigma(t) = \sum_{l=1}^L \tilde{w}^l (\mathbf{z}^l(t) - \mu(t))(\mathbf{z}^l(t) - \mu(t))^T$ are the mean and covariance of the state w.r.t. $\hat{p}^*$, respectively. Suppose a new set of trajectories and their weights is obtained by a (previous) control policy $\mathbf{u}(t) = \bar{\mathbf{u}}^{ff}(t) + \bar{\mathbf{K}}(t)\bar{\Sigma}^{-1/2}(t)(\mathbf{z}(t) - \bar{\mu}(t))$. Then, the path integral control theorem in Appendix B gives the update rules as:

$$\mathbf{u}^{ff}(t)dt = \bar{\mathbf{u}}^{ff}(t)dt + \bar{\mathbf{K}}(t)\bar{\Sigma}^{-1/2}(t)(\mu(t) - \bar{\mu}(t))dt + \eta \sum_{l=1}^L \tilde{w}^l d\mathbf{w}^l(t), \tag{15}$$

$$\mathbf{K}(t)dt = \bar{\mathbf{K}}(t)\bar{\Sigma}^{-1/2}(t)\Sigma^{1/2}(t)dt + \eta \sum_{l=1}^L \tilde{w}^l d\mathbf{w}^l(t)\left(\Sigma^{-1/2}(t)(\mathbf{z}^l(t) - \mu(t))\right)^T, \tag{16}$$

with the adaptation rate $\eta$. The initial state distribution also can be updated into $q_0(\cdot) = \mathcal{N}(\cdot; \hat{\mu}_0, \hat{\Sigma}_0)$:

$$\hat{\mu}_0 = \sum_{l=1}^L \tilde{w}^l \mathbf{z}^l(0), \ \hat{\Sigma}_0 = \sum_{l=1}^L \tilde{w}^l(\mathbf{z}^l(0) - \hat{\mu}_0)(\mathbf{z}^l(0) - \hat{\mu}_0)^T. \tag{17}$$

Starting from the variational parameters, $\{\hat{\mu}_0, \hat{\Sigma}_0, \mathbf{u}^{ff}_{1:K-1}, \mathbf{K}_{1:K-1}\}$, given by the inference network and $\bar{\mu}(t) = 0, \bar{\Sigma}(t) = I$, the update rules in (15)-(17) gradually refine the parameters of $q_{\mathbf{u}}$ in order for the resulting trajectory distribution to be close to the posterior distribution. After $R$ adaptations, the MCO and its gradient are estimated by:

$$\hat{\mathcal{L}}^L = \log \frac{1}{L} \sum_{l=1}^L \exp(-S_{\mathbf{u}}(\mathbf{z}^l_{[0,T]})), \ \nabla_{\theta,\phi}\hat{\mathcal{L}}^L = -\sum_{l=1}^L \tilde{w}^l \nabla_{\theta,\phi} S_{\mathbf{u}}(\mathbf{z}^l_{[0,T]}), \tag{18}$$

where $\theta$ and $\phi$ denote the parameters of the generative model, i.e., $\mathbf{f}(\mathbf{z}), \sigma(\mathbf{z}), p_0(\mathbf{z})$ and $p(\mathbf{x}|\mathbf{z})$, and the inference network, i.e., the backward RNN, respectively. Because all procedures in the path integral adaptation and MCO construction are differentiable, they can be implemented by a fully differentiable network with $R$ recurrences, which we named Adaptive Path Integral Autoencoder (APIAE); see also Fig. 3(b) in the Appendix C.

Note that the inference, reconstruction, and gradient backpropagation of APIAE can operate independently for each of $L$ samples. Consequently, the computational cost grows linearly with the number of samples, $L$, and the number of adaptations, $R$. As implemented in IWAE [Burda et al., 2016], we replicated each observation data $L$ times and the whole operations were parallelized with GPU. We implemented APIAE with Tensorflow [Abadi et al., 2016]; the pseudo code and algorithmic details of APIAE are given in the Appendix C.

## 4 High-dimensional Motion Planning with Learned Latent Model

High-dimensional motion planning is a challenging problem because of the curse of dimensionality: The size of the configuration space exponentially increases with the number of dimensions. However, like in the latent variable model learning, it might be a reasonable assumption that configurations a planning algorithm really needs to consider form some sort of low-dimensional manifold in the configuration space [Vernaza and Lee, 2012], and the learned generative model provides stochastic dynamics in that manifold. Once this low-dimensional representation is obtained, any motion planning algorithm can solve high-dimensional planning problem very efficiently by utilizing it to restrict the search space.

More formally, suppose that the initial configuration, $\mathbf{x}_1$, and corresponding latent state, $\mathbf{z}(0)$, are given and the cost function, $C_k(\mathbf{x}_k)$, encodes given task specifications of a planning problem, e.g.,

desirability/undesirability of certain configurations, a penalty for obstacle collision, etc. Then, the planning problem can be converted into the problem of finding the optimal trajectory distribution, $q_{\mathbf{u}}$, that minimizes the following objective function:

$$J(q_{\mathbf{u}}) = \mathbb{E}_{\mathbf{x}_{1:K} \sim p_\theta(\cdot|\mathbf{z}_{[0,T]}), \mathbf{z}_{[0,T]} \sim q_{\mathbf{u}}(\cdot)} \left[ \sum_{k=1}^{K} C_k(\mathbf{x}_k) + D_{KL}(q_{\mathbf{u}}(\mathbf{z}_{[0,T]}) || p_\theta(\mathbf{z}_{[0,T]})) \right]. \qquad (19)$$

That is, we want to find parameters, $\mathbf{u}$, of the trajectory distribution which not only is likely to generate sample configuration sequences achieving the lower planning cost but also does not deviate a lot from the (learned) prior, $p_\theta(\mathbf{z}_{[0,T]})$. The solution can be found using the aforementioned adaptive path integral control method, where its state cost function is set as: $V(\mathbf{z}_{[0,T]}) \equiv \mathbb{E}_{p_\theta(\mathbf{x}_{1:K}|\mathbf{z}_{[0,T]})} \left[ \sum_{k=1}^{K} C_k(\mathbf{x}_k) \right]$ and the initial state distribution is not updated in the adaptation process. After the adaptations with this state cost function, the resulting plan can simply be sampled from the generative model, e.g., $\mathbf{x}_{1:K} \sim p_\theta(\cdot|\mu_{[0,T]})$. Note that the time interval $t_k - t_{k-1}$ and the trajectory length $K$ can differ in the training and planning phases because continuous-time dynamics is dealt with.

## 5 Related Work

To address the complexity raised from temporal structures of data, several approaches that build a sophisticated approximate inference model have been proposed. For example, Karl et al. [2017] used the locally linear latent dynamics by introducing transition parameters, where an inference model infers transition parameters rather than latent states from the local transition. Johnson et al. [2016] combined a structured graphical model in latent space with a deep generative network, where an inference network produces local evidence potentials for the message passing algorithms. Fraccaro et al. [2017] constructed two layers of latent models, where linear-Gaussian dynamical systems governed two latent layers and the observation at each time step was related to the middle layer independently; the inference model in this framework consists of independent VAE's inference networks at each time-step and the Kalman smoothing algorithm along the time axis. Finally, deep Kalman smoother (DKS) in [Krishnan et al., 2017] parameterized the dynamical system by a deep neural network and built an inference network as it has the same structure with the factorized posterior distribution. The idea of MCOs was also used in the temporal setting. Maddison et al. [2017], Le et al. [2018], Naesseth et al. [2018] adapted the particle filter (PF) algorithm as their inference models and utilized a PF's estimator of the marginal likelihood as an objective function of training which Maddison et al. [2017] named the filtering variational objectives (FIVOs).

These approaches can be viewed as attempts to reduce the approximation gap; by building the inference model in sophisticated ways that exploit underlying structure of data, the resulting variational family could flexibly approximate the posterior distribution. To overcome the amortization gap caused by inference networks, the *semi-amortized* method utilizes an iterative refinement procedure for improving variational distribution. Let $q_\phi$ and $q^*$ be the variational distributions from the inference network and from the refinement procedure, i.e., before and after the refinement, respectively. Hjelm et al. [2016] adopted adaptive importance sampling to refine the variational parameters, and the generative and inference networks are trained separately with $\bigtriangledown_\theta \mathcal{L}(q^*, \theta; \mathbf{x})$ and $\bigtriangledown_\phi D_{KL}(q^*||q_\phi)$, respectively. Krishnan et al. [2018] used stochastic variational inference as a refinement procedure, and the generative and inference networks are also trained separately with $\bigtriangledown_\theta \mathcal{L}(q^*, \theta; \mathbf{x})$ and $\bigtriangledown_\phi \mathcal{L}(q_\phi, \theta; \mathbf{x})$, respectively. Kim et al. [2018] also used stochastic variational inference but proposed the *end-to-end* training by allowing the learning signals to be backpropagated into the refinement procedure, and showed this end-to-end training outperformed the separate training.

This work presents a semi-amortized variational inference method for temporal data. In summary, we parameterize the variational distribution by control input and transformed the approximate inference into the SOC problem. Our method utilizes the structured inference network based on the principle of optimality which has a similar structure to the inference network of DKS [Krishnan et al., 2017]. The adaptive path-integral control method, which can be viewed as adaptive importance sampling in trajectory space [Kappen and Ruiz, 2016], is then adopted as a refinement procedure. Ruiz and Kappen [2017] also used the adaptive path integral approach to solve smoothing problems and showed the path integral-based smoothing method could outperform the PF-based smoothing algorithms. Finally, by observing all procedures of the path integral smoothing are differentiable, the inference and generative networks are trained in the end-to-end manner. Note that APIAE is not the first

algorithm that implements an optimal planning/control algorithm into a fully-differentiable network. In [Tamar et al., 2016, Okada et al., 2017, Karkus et al., 2017], similar iterative refinement procedures were built as differentiable networks to learn solutions of *control problems* in an end-to-end manner; the fact that iterative methods were generally used to solve control problems can be a rationale for utilizing refinement to approximate inference for sequential data.

In addition, there is a *non-probabilistic* branch of representation learning of dynamical systems, e.g., [Watter et al., 2015, Banijamali et al., 2018, Jonschkowski and Brock, 2015, Lesort et al., 2018]. They basically stack two consecutive observations to contain the temporal information and learn the dynamical model based on a carefully designed loss function considering the stacked data as one observation. As shown in Appendix D, however, when the observations are highly-noisy (or even worse, when the system is unobservable with the stacked data), stacking a small number of observations prohibits the training data from containing enough temporal information for learning rich generative models.

Lastly, there have been some recent works to utilize a low-dimensional latent model for motion planning. Chen et al. [2016] exploited the idea of VAEs to embed dynamic movement primitives into the latent space. In [Ha et al., 2018], Gaussian process dynamical models [Wang et al., 2008] served as a latent dynamical model and was utilized for planning in a similar way with this work. Though the dynamics were not considered, Ichter et al. [2018], Zhang et al. [2018] used the conditional VAEs to learn a non-uniform sampling methodology of a sampling-based motion planning algorithm.

# 6 Experiment

In our experiments, we would like to show that the proposed method is a complementary technique to the existing methods; the APIAE can play a role in constructing more expressive posterior distribution by refining the variational distribution from the existing approximate inference methods. To support our statement, we built APIAEs upon the FIVO and IWAE frameworks and compared with the model without adaptation procedures.

We set our APIAE parameters as L=8, R=4, and K=10 during experiments. Quantitative studies about the effect of varying these parameters are discussed in the appendix. Feedback gain is only used for the planning, since matrix inversion in (16) requires Cholesky decomposition which is often numerically unstable during the training. We would refer the readers to the Appendix D and the supplementary video for more experimental details and results.

## 6.1 Dynamic Pendulum

The first experiment addresses the system identification and planning of inverted pendulum with the raw images. The pendulum dynamics is represented by the second order differential equation for angle of the pendulum, $\psi$, as $\ddot{\psi} = -9.8 \sin(\psi) - \dot{\psi}$. We simulated the pendulum dynamics by injecting the disturbance from random initial states and then made sequences of $16 \times 16$ sized images corresponding to the pendulum state with the time interval, $\delta t = 0.1$. This set of sequence images was training data of APIAE, i.e., $\mathbf{x}_k$ lied in 256-dimensional observation space. 3000 and 500 data are used for training and test, respectively.

Fig. 1(a) shows the constructed 2-dimensional latent space; each point represents the posterior mean of the observation data and it is shown that the angle and the angular velocity are well-encoded in 2-dimensional space. As shown in Fig. 1(b), the learned dynamical model was able to successfully reconstruct the noisy observations, predict and plan the future images. For the planning, the cost functions were set to penalize the difference between the last image of the generated sequence and the target image in Fig. 1(c) to encode planning problems for swing-up, -down, -left, and -right.

## 6.2 Human Motion Capture Data

The second experiment addresses a motion planning of a humanoid robot with 62-dimensional configuration space. We utilized human motion capture data from the Carnegie Mellon University motion capture (CMU mocap) database for the learning; the training data was a set of (short) locomotion, e.g., for standing, walking, and turning. The 62-dimensional configurations consist of angles of all joints, roll and pitch angles, vertical position of the root, yaw rate of the root,

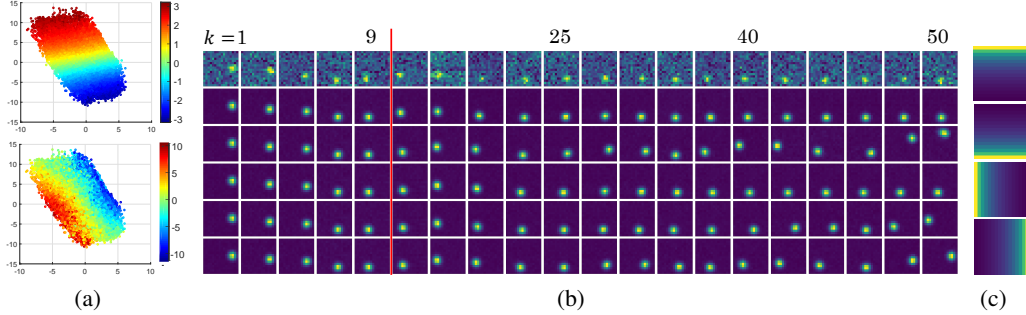

Figure 1: Pendulum results. (a) The inferred latent states colored by angles (top) and angular velocities (bottom) of the ground truth. (b) Resulting image sequences. From the top: images of ground truth, prediction, and four plaining results for swing-up, -down, -left, and -right, respectively. Except the first row, the images before the red line ($k \leq 10$) are reconstructed one. (c) The target images for each task: $C_K = ||\mathbf{x}_{target} - \mathbf{x}_K||^2$.

and horizontal velocity of the root. The global (horizontal) position and heading orientation are not encoded in the generative model (only velocities are encoded), but they can be recovered by integration when an observation sequence is given. The original data were written at 120 Hz, and we down-sampled them to 20 Hz and cut them every 10 time steps, i.e., $\delta t = 0.05$, $K = 10$. 1043 and 173 data are used for training and test, respectively. We utilized the DeepMind Control Suite [Tassa et al., 2018] for parsing the data and visualizing the results.

Figs. 2(a-c) illustrate the posterior mean states of the training data colored by some physical quantities of the ground truth; we can observe that (a) locomotion is basically embedded along the surface of the cylinder, while (b) they were arranged in the order of the yaw rates along the major axis of the cylinder and (c) motions with lower forward velocities were embedded into smaller radius cycles. Also, Fig. 2(d) shows that APIAE successfully reconstructed the data. Compared to the pendulum example, where the Wiener process in latent dynamics models disturbance into the system and the prediction can be made simply by ignoring the disturbance, the framework in this example uses the Wiener process to model the uncertainty in human's decision, e.g., whether to turn left or right, to increase or decrease their speed, etc, similar to the modeling of the bounded rationality [Genewein et al., 2015] or the maximum entropy IRL [Ziebart et al., 2008]; as shown in Fig. 2(e), from the very same initial pose, the framework predicts multiple future configurations for, e.g., going straight, turning left or right (the ratio between motions eventually matches that of the training dataset) and these predictions play essential roles in the planning. We then formulated planning problems, where the cost function penalized collision with an obstacle, large yaw rate, and distance from the goal. Figs. 2(f-g) show that the proposed method successfully generated the natural and collision-free motion toward the goal.

## 6.3 Quantitative Results

It is easily thought that powerful inference methods via resampling or refinements make the bound tighter, but achieving a tighter bound during learning does not directly imply a better model learning [Rainforth et al., 2018]. To investigate this, we have compared the lower bound, the reconstruction and prediction abilities of the models learned by the proposed and baseline algorithms. The results are reported in Table 1 (higher is better).[4] Interestingly, we can observe that learning with both the resampling and path-integral refinements resulted in the best reconstruction ability as well as the tightest bound, but the best prediction was achieved by the model learned only with the refinements. It implies that while powerful inference can lead to a tighter bound and a good reconstruction, a bias in the gradients can prevent the resulting model from being accurate (note that the gradient components from the resampling are generally ignored because it causes high variance of the gradient estimator [Maddison et al., 2017, Le et al., 2018, Naesseth et al., 2018]). In the planning side, the prediction power is crucial because the (learned) generative model needs to sample meaningful and diverse configuration sequences. We conclude that the resampling procedure would be better to utilize only

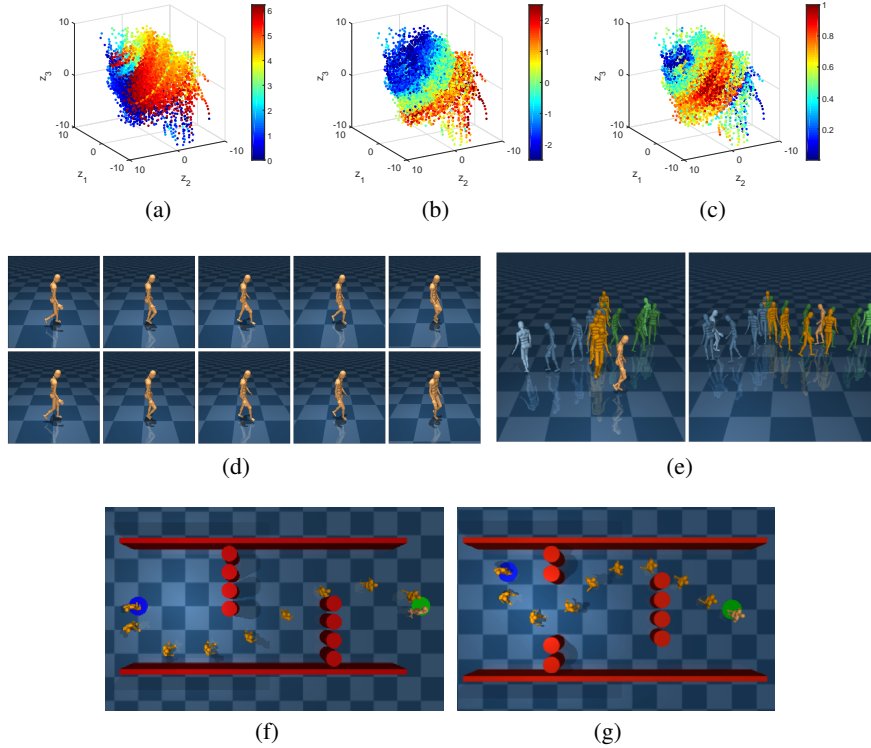

Figure 2: Mocap results. The learned latent space colored by (a) the gait phase, (b) yaw rate, and (c) forward velocity of the ground truth. We set the phase as 0 when the left foot touch the ground and as $\pi$ when the right foot touch the ground. (d) Reconstruction. (e) Prediction results from the same initial poses. (f-g) Locomotion planning results.

for planning, not for learning, and this also would be the same in other application domains like 3-dimensional human motion tracking, where the prediction ability is more important.

Table 1: Comparison of the lower bound, reconstruction, and prediction. Each model was trained with (i) APIAE with resampling (+r), (ii) APIAE without resampling, (iii) FIVO, and (iv) IWAE. The lower bounds are obtained for the training datasets and the reconstruction and prediction results are made for the test datasets; the amounts of the test datasets were around 1/6 of the training datasets.

|  | Pendulum ($\times 10^6$) | | | Mocap ($\times 10^5$) | |
|---|---|---|---|---|---|
|  | Lower-bound | Reconstruction | Prediction | Lower-bound | Reconstruction |
| APIAE+r | **-9.866** | **-1.647** | -1.985 | **-6.665** | **-1.158** |
| APIAE | -9.927 | -1.653 | **-1.845** | -6.680 | -1.171 |
| FIVO | -9.890 | -1.650 | -1.978 | -6.687 | -1.167 |
| IWAE | -9.974 | -1.665 | -1.860 | -6.683 | -1.174 |

# 7   Conclusion

In this paper, a semi-amortized variational inference method for sequential data was proposed. We parameterized a variational distribution by control input and transformed an approximate inference into a SOC problem. The proposed framework utilized the structured inference network based on the principle of optimality and adopted the adaptive path-integral control method as a refinement procedure. The experiments showed that the refinement procedure helped the learning algorithm achieve tighter lower bound. Also, it is shown that the valid dynamical model can be identified from sequential raw data and utilized to plan the future configurations.

**Acknowledgments**

This work was supported by the Agency for Defense Development under contract UD150047JD.

## Footnotes

[1] https://youtu.be/xCp35crUoLQ

[2] https://github.com/yjparkLiCS/18-NeurIPS-APIAE

[3]Because each observation trajectory can be considered independently, we leave trajectory index, $i$, out and restrict our discussion to one trajectory for the sake of notational simplicity.

[4]Mocap prediction is omitted, because a proper measure for the prediction is unclear.

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
