[Supplementary Material]

# Appendix

*[Supplementary material for J.-S. Ha, Y.-J. Park, H.-J. Chae, S.-S. Park, and H.-L. Choi, "Adaptive Path-Integral Autoencoder: Representation Learning and Planning for Dynamical Systems," NeurIPS 2018.]*

## A Objective Function of Linearly-Solvable Optimal Control

Suppose that an objective function of a SOC problem is given as:

$$J = \mathbb{E}_{q_{\mathbf{u}}} \left[ \int_0^T V(\mathbf{z}(t)) + \frac{1}{2} ||\mathbf{u}(t)||^2 dt \right] + D_{KL} \left( q_0(\mathbf{z}(0)) || p_0(\mathbf{z}(0)) \right), \tag{20}$$

where $q_{\mathbf{u}}$ is the probability measures induced by the controlled trajectories from (9). The first and second terms in the integral encodes a state cost and regulates control input effort, respectively, and the last KL term penalizes the initial state deviation. The objective of the SOC problem is to find the optimal control sequence $\mathbf{u}^*(t)$ as well as the initial state distribution $q_0$, with which the trajectory distribution of (9) minimizes the objective function (20).

The following theorem implies that the control penalty term in (20) can be interpreted as the KL-divergence between distributions of controlled and uncontrolled trajectories.

**Theorem 1 (Girsanov's Theorem (modified from Gardiner et al. [1985]))** *Suppose $p$ and $q_{\mathbf{u}}$ are the probability measures induced by the trajectories of* (3) *and* (9)*, respectively. Then, the Radon-Nikodym derivative of $q_{\mathbf{u}}$ with respect to $p$ is given by*

$$\frac{dp(\mathbf{z}_{[0,T]})}{dq_{\mathbf{u}}(\mathbf{z}_{[0,T]})} = \frac{p_0(\mathbf{z}(0))}{q_0(\mathbf{z}(0))} \exp \left( -\frac{1}{2} \int_0^T ||\mathbf{u}(t)||^2 dt - \int_0^T \mathbf{u}(t)^T d\mathbf{w}(t) \right), \tag{21}$$

*where $\mathbf{w}(t)$ is a Wiener process for simulating $q_{\mathbf{u}}$.*

With Girsanov's theorem, the objective function (20) is rewritten in the form of the KL-divergence:

$$\begin{aligned}
J &= \mathbb{E}_{q_{\mathbf{u}}} \left[ \int_0^T V(\mathbf{z}(t)) + \frac{1}{2} ||\mathbf{u}(t)||^2 dt \right] + D_{KL} \left( q_0(\mathbf{z}(0)) || p_0(\mathbf{z}(0)) \right) \\
&= \mathbb{E}_{q_{\mathbf{u}}} \left[ \int_0^T V(\mathbf{z}(t)) dt + \log \frac{dq_{\mathbf{u}}(\mathbf{z}_{[0,T]})}{dp(\mathbf{z}_{[0,T]})} - \log \frac{q_0(\mathbf{z}(0))}{p_0(\mathbf{z}(0))} \right] + D_{KL} \left( q_0(\mathbf{z}(0)) || p_0(\mathbf{z}(0)) \right) \\
&= \mathbb{E}_{q_{\mathbf{u}}} \left[ \log \frac{dq_{\mathbf{u}}(\mathbf{z}_{[0,T]})}{dp(\mathbf{z}_{[0,T]}) \exp(-V(\mathbf{z}_{[0,T]}))/\xi} - \log \xi \right] \\
&= D_{KL} \left( q_{\mathbf{u}}(\mathbf{z}_{[0,T]}) || p^*(\mathbf{z}_{[0,T]}) \right) - \log \xi, \tag{22}
\end{aligned}$$

where $V(\mathbf{z}_{[0,T]}) \equiv \int_0^T V(\mathbf{z}(t)) dt$ is a trajectory state cost and $\xi \equiv \int \exp(-V(\mathbf{z}_{[0,T]})) dp(\mathbf{z}_{[0,T]})$ is a normalization constant. Note that the second term in the exponent of (21) disappears when taking expectation w.r.t. $q_{\mathbf{u}}$, i.e. $\mathbb{E}_{q_{\mathbf{u}}}[\int_0^T \mathbf{u}(t)^T d\mathbf{w}(t)] = 0$, because $\mathbf{w}(t)$ is a Wiener process for simulating $q_{\mathbf{u}}$. Because $\xi$ is not a function of $\mathbf{u}$, $p^*(\mathbf{z}_{[0,T]})$ can be interpreted as the optimally-controlled trajectory distribution that minimizes the objective function, $J$:

$$dp^*(\mathbf{z}_{[0,T]}) = \frac{\exp(-V(\mathbf{z}_{[0,T]})) dp(\mathbf{z}_{[0,T]})}{\int \exp(-V(\mathbf{z}_{[0,T]})) dp(\mathbf{z}_{[0,T]})} \tag{23}$$

$$\propto dp(\mathbf{z}_{[0,T]}) \exp(-V(\mathbf{z}_{[0,T]})). \tag{24}$$

This expression yields the method to sample the optimally-controlled trajectories: We first sample a set of trajectories according to the passive dynamics, i.e., $\mathbf{z}_{[0,T]}^l \sim p(\cdot)$, which can be interpreted as the proposal distribution, and assign their importance weights as $\tilde{w}^l \propto \exp(-V(\mathbf{z}_{[0,T]}^l))$, $\forall l$ and $\sum_l \tilde{w}^l = 1$.

The proposal distribution can be changed into the controlled trajectory distribution so as to increase the sample efficiency. By applying the Girsanov's theorem again, the optimal trajectory distribution is expressed as:

$$dp^*(\mathbf{z}_{[0,T]}) \propto dq_{\mathbf{u}}(\mathbf{z}_{[0,T]}) \exp\left(-S_{\mathbf{u}}(\mathbf{z}_{[0,T]})\right), \tag{25}$$

where

$$S_{\mathbf{u}}(\mathbf{z}_{[0,T]}) = V(\mathbf{z}_{[0,T]}) + \frac{1}{2}\int_0^T ||\mathbf{u}(t)||^2 dt + \int_0^T \mathbf{u}(t)'d\mathbf{w}(t). \tag{26}$$

This yields that the optimal trajectory distribution can be obtained by sampling a set of trajectories according to the controlled dynamics with $\mathbf{u}(t)$, i.e., $\mathbf{z}_{[0,T]}^l \sim q_{\mathbf{u}}(\cdot)$, and assigning their importance weights as $\tilde{w}^l \propto \exp(-S_{\mathbf{u}}(\mathbf{z}_{[0,T]}^l)), \forall l$ and $\sum_l \tilde{w}^l = 1$. It is known that, as the control input $\mathbf{u}(\cdot)$ gets closer to the true optimal control input $\mathbf{u}^*(\cdot)$, the variance of importance weights decreases and it reduces to 0 when $\mathbf{u}(t) = \mathbf{u}^*(t, \mathbf{z}(t))$ [Thijssen and Kappen, 2015].

## B   Derivation of Path Integral Adaptation

From the trajectories sampled with $q_{\mathbf{u}}(\cdot)$, the path integral control provides how to compute the optimal control $\mathbf{u}^*(t)$ based on the following theorem.

**Theorem 2 (Main Theorem [Thijssen and Kappen, 2015])** *Let* $f : \mathbb{R} \times \mathbb{R}^{d_z} \to \mathbb{R}$, *and consider the process* $f(t) = f(t, \mathbf{z}(t))$ *with* $\mathbf{z}_{[0,T]} \sim q_{\mathbf{u}}(\cdot)$. *Then,*

$$\langle (\mathbf{u}^* - \mathbf{u})f \rangle (t) = \lim_{\tau \to t} \left\langle \frac{\int_t^\tau f(s)d\mathbf{w}(s)}{\tau - t} \right\rangle, \tag{27}$$

*where* $\langle Y(t) \rangle \equiv \mathbb{E}_{q_{\mathbf{u}}}[\tilde{w}_{\mathbf{u}} Y(t)]$, $\tilde{w}_{\mathbf{u}} = \frac{\exp(-S_{\mathbf{u}}(\mathbf{z}_{[0,T]}))}{\mathbb{E}_{q_{\mathbf{u}}}[\exp(-S_{\mathbf{u}}(\mathbf{z}_{[0,T]}))]}$ *for any process* $Y(t)$.

Suppose the current control policy is parameterized with $n_b$ basis functions $\bar{h}(t, \mathbf{z}) : \mathbb{R} \times \mathbb{R}^{d_z} \to \mathbb{R}^{n_b}$ as:

$$\bar{\mathbf{u}}(t, \mathbf{z}(t)) = \bar{\mathbf{A}}(t)\bar{h}(t, \mathbf{z}(t)), \tag{28}$$

where $\bar{\mathbf{A}}(t) : \mathbb{R} \to \mathbb{R}^{d_u \times n_b}$ is the control policy parameter and let the optimal parameterized control policy be $\mathbf{u}^* = \mathbf{A}^*(t)h(t, \mathbf{z}(t))$. Then, Theorem 2 can be rewritten as:

$$\mathbf{A}^*(t) \langle h \otimes h \rangle (t) = \bar{\mathbf{A}}(t) \langle \bar{h} \otimes h \rangle (t) + \lim_{\tau \to t} \left\langle \frac{\int_t^\tau d\mathbf{w}(s) \otimes h(s)}{\tau - t} \right\rangle. \tag{29}$$

Because we can utilize only a finite number of samples to approximate the optimal trajectory distribution, it is more reasonable to update the control policy parameter with some small adaptation rate, than to estimate it at once. Similar to Ruiz and Kappen [2017], we use a standardized linear feedback controller w.r.t. the target distribution, i.e.,

$$h(t, \mathbf{z}(t)) \equiv \left[1; \Sigma^{-1/2}(t)(\mathbf{z}(t) - \mu(t))\right], \tag{30}$$

where $\mu(t) = \langle \mathbf{z}(t) \rangle$ and $\Sigma(t) = \langle (\mathbf{z}(t) - \mu(t))(\mathbf{z}(t) - \mu(t))^T \rangle$ are the mean and covariance of the state w.r.t. the optimal trajectory distribution estimated at the previous iteration. Then, the control input has a form as:

$$\mathbf{u}(t) = \mathbf{u}^{ff}(t) + \mathbf{K}(t)\Sigma^{-1/2}(t)(\mathbf{z}(t) - \mu(t)), \tag{31}$$

where the parameter, $\mathbf{A}(t) = [\mathbf{u}^{ff}(t), \mathbf{K}(t)]$, represents feedforward control signal and feedback gain.

Suppose we have a set of trajectories and their weights obtained by the parameterized policy, $\bar{\mathbf{u}}(t) = \bar{\mathbf{A}}(t)\bar{h}(t, \mathbf{z}(t))$. Then, based on (29), the control policy parameters can be updated as follows:

$$\mathbf{u}^{ff}(t)dt = \bar{\mathbf{u}}^{ff}(t)dt + \bar{\mathbf{K}}(t)\bar{\Sigma}^{-1/2}(t)(\mu(t) - \bar{\mu}(t))dt + \eta \langle d\mathbf{w}(t) \rangle, \tag{32}$$

$$\mathbf{K}(t)dt = \bar{\mathbf{K}}(t)\bar{\Sigma}^{-1/2}(t)\Sigma^{1/2}(t)dt + \eta \left\langle d\mathbf{w}(t) \left(\Sigma^{-1/2}(t)(\mathbf{z}(t) - \mu(t))\right)^T \right\rangle, \tag{33}$$

(a) Structured inference network         (b) APIAE

Figure 3: Overall structure of the proposed method

where $\eta$ is an adaptation rate[5]. Note that the adaptation of two terms can be done independently, because $\langle h \otimes h \rangle (t) = I$. Beside the control policy adaptation, the initial state distribution, $p_0$, can be updated as well:

$$\hat{\mu}_0 = \langle \mathbf{z}(0) \rangle, \ \hat{\Sigma}_0 = \left\langle (\mathbf{z}(0) - \hat{\mu}_0)(\mathbf{z}(0) - \hat{\mu}_0)^T \right\rangle, \tag{34}$$

where the updated trajectory distribution starts from $q_0(\cdot) = \mathcal{N}(\cdot; \hat{\mu}_0, \hat{\Sigma}_0)$. The whole procedures are summarized in Algorithm 1.

---

**Algorithm 1** Path Integral Adaptation

---

**Input**: Dynamics, $\mathbf{f}(\mathbf{z})$, $\sigma(\mathbf{z})$, initial state distribution, $\hat{\mu}_0, \hat{\Sigma}_0$, and control policy parameters, $\mathbf{A}_{[0,T]}$.

1: **for** $r \in \{1, ..., R\}$ **do**
2:      $\{S_{\mathbf{u}}, \hat{w}, \mathbf{z}_{[0,T]}, \mathbf{w}_{[0,T]}\}^{1:L} \leftarrow \text{SIMULATE}(\hat{\mu}_0, \hat{\Sigma}_0, \mathbf{A}_{[0,T]})$
3:      $\hat{\mu}_0, \hat{\Sigma}_0, \mathbf{A}_{[0,T]} \leftarrow \text{IMPROVE}(\{\hat{w}, \mathbf{z}_{[0,T]}, \mathbf{w}_{[0,T]}\}^{1:L}, \mathbf{A}_{[0,T]})$        ▷ using (15)-(16) and (17)
4: **end for**
5: $\{S_{\mathbf{u}}, \mathbf{z}_{[0,T]}, \mathbf{w}_{[0,T]}\}^{1:L} \leftarrow \text{SIMULATE}(\hat{\mu}_0, \hat{\Sigma}_0, \mathbf{A}_{[0,T]})$
6: **return** $\{\mathbf{z}_{[0,T]}, S_{\mathbf{u}}, \hat{w}\}^{1:L}$

---

1: **function** $\text{SIMULATE}(\hat{\mu}_0, \hat{\Sigma}_0, \mathbf{A}_{[0,T]})$           ▷ Stochastic simulation via Euler method
2:      $\mathbf{z}_1^{1:L} \leftarrow \text{SAMPLENORMAL}(\hat{\mu}_0, \hat{\Sigma}_0)$
3:      **for** $k \in \{1, ..., K-1\}$ **do**
4:          **for** $l \in \{1, ..., L\}$ **do**
5:              $d\mathbf{w}_{k-1}^{(l)} \leftarrow \text{SAMPLENORMAL}(0, \sqrt{\delta t}I)$
6:              $\mathbf{z}_k^{(l)} \leftarrow \mathbf{z}_{k-1}^{(l)} + \mathbf{f}(\mathbf{z}_{k-1}^{(l)})\delta t + \sigma(\mathbf{z}_{k-1}^{(l)})(\mathbf{u}_{k-1}^{(l)}\delta t + d\mathbf{w}_{k-1}^{(l)})$      ▷ $\mathbf{u}_{k-1}^{(l)}$ from (28).
7:              $S_{\mathbf{u}}^{(l)} \leftarrow S_{\mathbf{u}}^{(l)} + V(\mathbf{z}_k^{(l)})\delta t + \frac{1}{2}||\mathbf{u}_{k-1}^{(l)}||^2 \delta t + (\mathbf{u}_{k-1}^{(l)})^T d\mathbf{w}_{k-1}^{(l)}$
8:              $\hat{w}^{1:L} \leftarrow \exp(-S_{\mathbf{u}}^{1:L})/\sum_l \exp(-S_{\mathbf{u}}^l)$
9:              (Optional) Resample if effective sample size of $\hat{w}^{1:L}$ is smaller than threshold
10:          **end for**
11:      **end for**
12:      **return** $\{S_{\mathbf{u}}, \hat{w}, \mathbf{z}_{[0,T]}, \mathbf{w}_{[0,T]}\}^{1:L}$
13: **end function**

---

## C   Algorithmic Details

The pseudo code of APIAE training is shown in Algorithm 3. Given the observation data, the inference network implemented by the backward RNN first approximates the posterior distribution using Algorithm 2 (line 2–3). Then, the algorithm iteratively refines the variational distribution using path integral adaptation method in Algorithm 1 (line 4), estimates the lower bound of data likelihood

**Algorithm 2** Structured inference network $h_\phi$ (Figure 3(a))

---

**Input**: A observation sequence, $\mathbf{x}_{1:K}$.

1: $\mathbf{h}_K \leftarrow h_{\phi,r}(0, \mathbf{x}_K)$
2: **for** $k \in \{K-1, ..., 1\}$ **do**
3:     $\mathbf{h}_k \leftarrow h_{\phi,r}(\mathbf{h}_{k+1}, \mathbf{x}_k)$                            $\triangleright$ recurrence
4:     $\mathbf{A}_k \leftarrow h_{\phi,o_1}(\mathbf{h}_k, \mathbf{h}_{k+1})$                           $\triangleright$ output
5: **end for**
6: $\{\hat{\mu}_0, \hat{\Sigma}_0\} \leftarrow h_{\phi,o_2}(\mathbf{h}_1)$                                     $\triangleright$ output
7: **return** $\{\hat{\mu}_0, \hat{\Sigma}_0, \mathbf{A}_{[0,T]}\}$

---

**Algorithm 3** Training of Adaptive Path Integral Autoencoder (Figure 3(b))

---

**Input**: Dataset of observation sequences, $\mathcal{D} = \{\mathbf{x}_{1:K}^{(i)}\}_{i=1,...,N}$.
           Latent and observation models, $\mathbf{f}(\mathbf{z}), \sigma(\mathbf{z}), p_0(\mathbf{z})$ and $p(\mathbf{x}|\mathbf{z})$, parameterized by $\theta$.
           Backward RNN as an inference network $h_\phi : \mathbf{x}_{1:K} \rightarrow \{\hat{\mu}_0, \hat{\Sigma}_0, \mathbf{A}_{[0,T]}\}$, parameterized by $\phi$.

1: **while** $notConverged()$ **do**
2:     Sample datapoint $\mathbf{x}_{1:K}$ from $\mathcal{D}$
3:     Initialize $\{\hat{\mu}_0, \hat{\Sigma}_0, \mathbf{A}_{[0,T]}\} \leftarrow h_\phi(\mathbf{x}_{1:K})$                 $\triangleright$ Algorithm 2
4:     $\{\mathbf{z}_{[0,T]}, S_\mathbf{u}, \hat{w}\}^{1:L} \leftarrow$ PI-ADAPTATION$(\hat{\mu}_0, \hat{\Sigma}_0, \mathbf{A}_{[0,T]}, \mathbf{x}_{1:K})$      $\triangleright$ Algorithm 1
5:     $\hat{\mathcal{L}} = \log \frac{1}{L} \sum_l \exp(-S_\mathbf{u}^{(l)})$, $\nabla_{(\theta,\phi)}\hat{\mathcal{L}} \leftarrow -\sum_l \hat{w}^{(l)} \nabla_{(\theta,\phi)} S_\mathbf{u}^{(l)}$
6:     Update $\theta$ and $\phi$ with $\nabla_{(\theta,\phi)}\hat{\mathcal{L}}$ using SGD     $\triangleright$ gradients are aggregated across mini-batches.
7: **end while**

---

and its gradients (line 5), and updates the model parameter according to the MCO gradients (line 6). The path integral adaptation and the MCO construction steps of APIAE can be seen as encoding and decoding procedures of autoencoders, respectively, motivating the name "adaptive path integral autoencoder."

## D  Experimental Details

### D.1  Pendulum

The latent space was set to be 2-dimensional and a locally-linear transition model used in Watter et al. [2015], Karl et al. [2017] were adopted, where the system dynamics were represented by combination of 16 linear systems as $\mathbf{f}_\theta = \sum_{i=1}^{16} \alpha^{(i)}(A^{(i)}\mathbf{z} + c^{(i)})$, $\sigma = \sum_{i=1}^{16} \alpha^{(i)} B^{(i)}$ and $\alpha = f_\lambda(\mathbf{z}) \in \mathbb{R}^{16}$ was a single layer neural network having 16 softmax outputs parameterized by $\lambda$, i.e., $\{A^{(i)}, B^{(i)}, c^{(i)}, \lambda\} \subset \theta$. For the stochastic simulation, we simply chose $t_k = (k-1)\delta t$, and $\delta t = T/(K-1)$. For the observation model, we considered a neural network with Gaussian outputs as $p(\cdot|\mathbf{z}) = \mathcal{N}(\cdot; \mu_\theta(\mathbf{z}), \sigma_\theta(\mathbf{z}))$, where $\mu_\theta(\cdot)$ and $\sigma_\theta(\cdot)$ are outputs of a neural network having 1a single hidden layer of 128 hidden units with ReLU activation and a $2 \times 256$-dimensional output layer without activation. We found that initializing dynamics network as stable results in the more stable learning, so the dynamics network was initialized with the supervised learning with transition data from stable linear system.

### D.2  Human Motion Capture Data

A 3-dimensional latent state space was used in this example and the dynamics were parameterized by the locally-linear transition model as in the pendulum experiment. The system dynamics were represented by combination of 16 linear systems as $\mathbf{f}_\theta = \sum_{i=1}^{16} \alpha^{(i)}(A^{(i)}\mathbf{z} + c^{(i)})$, $\sigma = \sum_{i=1}^{16} \alpha^{(i)} B^{(i)}$ and $\alpha = f_\lambda(\mathbf{z}) \in \mathbb{R}^{16}$ is a single layer neural network having 16 softmax outputs parameterized by $\lambda$, i.e., $\{A^{(i)}, B^{(i)}, c^{(i)}, \lambda\} \subset \theta$. For the stochastic simulation, we simply chose $t_k = (k-1)\delta t$, and $\delta t = T/(K-1)$. For the observation model, we considered a neural network with Gaussian outputs as: $p(\cdot|\mathbf{z}) = \mathcal{N}(\cdot; g_\theta(\mathbf{z}), I_{62})$, where $g_\theta(\cdot)$ is a neural network having a single hidden layer of 128 hidden units with ReLU activation and a 62-dimensional output layer without activation.

Table 2: The lower bound of log-likelihood for models trained with APIAEs w.r.t. the sample size.

| | Pendulum ($\times 10^6$) | | | | Mocap ($\times 10^5$) | | | |
|---|---|---|---|---|---|---|---|---|
| | L=4 | L=8 | L=16 | L=64 | L=4 | L=8 | L=16 | L=64 |
| APIAE+r | -9.9282 | -9.8380 | -9.8322 | -9.8306 | -6.689 | -6.665 | -6.637 | -6.683 |
| APIAE | -9.9724 | -9.9318 | -9.9153 | -9.8552 | -6.689 | -6.680 | -6.661 | -6.629 |

Table 3: The lower bound of data log-likelihood for models trained with APIAEs w.r.t. the number of path-integral adaptations.

| | Pendulum ($\times 10^6$) | | | Mocap ($\times 10^5$) | | |
|---|---|---|---|---|---|---|
| | R=0 | R=4 | R=8 | R=0 | R=4 | R=8 |
| APIAE+r | -9.890 | -9.866 | -9.795 | -6.687 | -6.665 | -6.648 |
| APIAE | -9.974 | -9.927 | -9.929 | -6.683 | -6.680 | -6.669 |

## D.3 Additional Results

To investigate the optimal parameters for APIAE training, we varied parameters of APIAEs, i.e., $L, \ R, \ K$, and compared the results.

Table 2 and Table 3 show the lower bounds of APIAEs for two experiments by varying the number of samples $L$ and adaptation $R$, respectively. As shown in the result, higher lower bound is achieved as the number of samples and adaptation get larger. Note, however, that APIAEs become computationally expensive as those parameters increase and slow down the training speed. Thus, we need to look for the compromise between the training efficiency and the performance. Empirically found that $L = 8$ and $R = 4$ show a reasonable performance with computational efficiency.

Fig. 4 show the learning results for the dataset of difference time length. It is observed that, when the observations are highly-noisy, the learning algorithm fails to extract enough temporal information from the data and then fails to build a valid generative dynamical model.

The lower bound of learned models are reported in Table 4 and Fig. 5. As in the Table 4, the highest lower bounds were achieved by the APIAE algorithms. Thus, the learning performances are seen to be improved via adaptation procedures with training on any bound. We also found that APIAEs produce higher bound than FIVO or IWAE throughout the training stage as shown in the Fig. 5.

Finally, Fig. 6 depicts the additional results of the Mocap experiment for the learned latent space, reconstruction, and prediction.

Table 4: Comparison of APIAE, FIVO, and IWAE bounds in the pendulum experiment. Each model was trained with (i) APIAE with resampling (+r), (ii) APIAE without resampling, (iii) FIVO, and (iv) IWAE. The resulting APIAE, FIVO, and IWAE bounds are shown.

| | Pendulum ($\times 10^6$) | | | | Mocap ($\times 10^5$) | | | |
|---|---|---|---|---|---|---|---|---|
| | APIAE+r | APIAE | FIVO | IWAE | APIAE+r | APIAE | FIVO | IWAE |
| APIAE+r | **-9.866** | -10.213 | -9.902 | -10.308 | **-6.665** | -6.694 | -6.683 | -6.723 |
| APIAE | -10.020 | **-9.927** | -10.037 | -9.953 | -6.712 | **-6.680** | -6.739 | -6.707 |
| FIVO | **-9.868** | -10.145 | -9.890 | -10.197 | **-6.675** | -6.691 | -6.687 | -6.711 |
| IWAE | -9.998 | **-9.959** | -10.145 | -9.974 | -6.694 | **-6.668** | -6.706 | -6.683 |

Figure 4: Pendulum experiment. The learned latent space colored by (top) angles and (bottom) angular velocities of the ground truth for different dataset with varying length, $K = 1, 2, 5, 10$.

Figure 5: Comparison of APIAE, FIVO, and IWAE bounds in the pendulum experiment. For each model trained with (a) APIAE with resampling, (b) APIAE without resampling, (c) FIVO, and (d) IWAE, the APIAE, FIVO, and IWAE bounds are shown.

Figure 6: (a-b) Locomotion reconstruction results. Top: ground truth, Bottom: reconstruction. (c) Prediction results from the same initial pose.

## Footnotes

[5]At the first iteration, $\bar{\mathbf{u}}^{ff}(t)$, $\bar{\mathbf{K}}(t)$ and $q_0$ are obtained from the inference network and $\bar{\mu}(t) = 0, \bar{\Sigma}(t) = I$.