[Reviews · NeurIPS 2018]

Reviewer 1



I had a hard time understanding the contributions of this paper. This might be partially due to the partial use of SDE notation that I’m not very familiar with, but it is also due to not very clear writing. I’m unsure about the originality of the content, and I am not convinced by the experiments (if I understand correctly what’s going on). I’m assigning a high degree of uncertainty to my score, as it could easily be that I’m missing some essential contributions. Detailed comments: How does the proposed approach differ from the prior work of [1,2] etc? Iterative refinement as been proposed as eg Posterior Policy Iteration (as has been in SOC). Just using a continuous time representation (that get discretized later anyways) would not convince me of sufficient novelty. Do the authors want to emphasize that they backpropagate through the refinement? I do not understand the Mocap experiments. Do they use the inferred controls to actually control the mujoco simulator with humanoid body? Also, I do not know how diverse the data set is. Are there test planning problem that are not in the training data. What is the model that is used? The appendix mentions a locally linear model. How is it parameterized exactly? I am unhappy about the introduction of the model in section 3.1. What’s the prior over z exactly? Is it the uncontrolled (ie u=0) version of the SDE from eqn 9? The videos of the pendulum problem confuse me. What does the left panel show during the swing-up task? If I see it correctly, the rendering on the right does a swing-up, but not the “ground truth” on the right. [1] On stochastic optimal control and reinforcement learning by approximate inference. Rawlik et al [2] Learning neural network policies with guided policy search under unknown dynamics. Levine, Abbeel.

Reviewer 2



Paper : 5353 This paper discusses representation learning problem, and designs an algorithm to learn variational parameters of approximate inference. The paper has adopted a semi-amortized technique for learning variational parameters of dynamic system for temporal data. In this paper, dynamic system is generated by state space model. In order to express the state space model, just initial state and control input at every time steps needs to be inferred. Paper has utilized adaptive path-integral technique for variational parameter refinement of dynamic system to mitigate the amortization gap( induced by limitation of inference network parameters compared to optimal variational distribution ). Expressing the problem(finding variational distribution parameters ) in the format that is addressed by stochastic optimal control and adaptation of adaptive path-integral technique to dynamic system can be considered main contributions of this paper . Experiments has been done on inverted pendulum and 3D human motion data, results show some small relative advantage for current method than baselines. For one of datasets (inverted pendulum), the model has shown good predictive capability although not compared with other baseline for quantitative prediction power. Quality: claims and supporting material both in main paper and supplementary have been verified and raised a few questions that. One issue found is in equation 10 appeared on page 4, as we know ELBO = E_{q_u(Z)}[log p(X|Z)] -KL(q_u(Z|X)||p(Z)), in supplementary materials shown that KL(q_u(Z|X)||p(Z)) = E_{q_u}[\integ{V(Z)+1/2 ||u(t)||^2}] +log \zetta where log\zetta does not contain the variational parameters, then ELBO will turn out to be ELBO = E_{q_u(Z)}[log p(X|Z) +\integ{-V(Z)-1/2 ||u(t)||^2}] , although what we have in 10 does not match with what I found in ELBO above, can you please clarify? Also I am not sure if V(z) in line 129 is same as V(z) in line 370 or just used same notation for 2 different expression. In experimental results of this paper,in inverted pendulum example, the prediction has been good and shows that parameters of variational distribution describes the dynamic system well. It will be informative if prediction for 3D human motion data is presented as well. Clarity: Although paper has been motivated well and general concepts and background have been explained well, there are some points which need clarification. On lines 137 and 138 has mentioned that feed forward/feedback structure can be implemented using RNN, but RNN has not been used in this paper for inferring parameters, please be more specific on that. Having schematic of network and specifying inference vs generative network and their parameters help the reader to understand your proposed solutions better. In experiment section baseline algorithms specification and setups are not mentioned Originality: As mentioned in summary the main contribution of this paper could be summarized as bellow - Expressing the problem(finding variational distribution parameters for modeling sequential data ) in the format that is addressed by stochastic optimal control (SOC) - adaptation of adaptive path-integral technique for variational parameter refinement of dynamic system to mitigate the amortization gap authors have cited many of recent works, and seems have done an in depth literature review also could distinguish their contribution to what has been done so far clearly. Significance: As discussed, in summary section, the results show some small relative improvement with baseline algorithms. But as suggested, it will be very useful to have a quantitative prediction comparison between the proposed algorithm and baselines.

Reviewer 3



This paper describes a method to learn a low-dimensional dynamical system from high-dimensional observation data. The method builds upon the so-called amortized inference, more precisely on semi-amortized inference with iterative refinement. Thanks to the SOC formalization, the auto-encoder is learning the low-dimensional sequence of states from the high-dimensional data. The manuscript reports resulst on the inverted pendulum application as well as real motion capture data. The reported results demonstrate that the proposed approach reaches a tighter bound. I am unable to double-check every single mathematical derivation done through the paper, since they are quite a few and some of them very technical. From an experimental perspective, I think having a "toy" and a "real" dataset is a nice choice. I am most concerned, however, on the pratical implications of the study. I understand the importance of estimating a tighter bound, but it is equally important to understand the practical effects of this tighter bound. Is it really meaningful in practice? Can we get better performance in some applicative scenario? These considerations are important, and to my understanding, missing in the paper.